# Predicting primate tongue morphology based on geometrical skull matching. A first step towards an application on fossil hominins

**Pablo Alvarez**[1,2], **Marouane El Mouss**[1], **Maxime Calka**[1], **Anca Belme**[1,3], **Gilles Berillon**[4], **Pauline Brige**[5], **Yohan Payan**[2], **Pascal Perrier**[6], **Amélie Vialet**[1,4] *

**1** Sorbonne Université, Institut des Sciences du Calcul et des Données, Paris, France, **2** Univ. Grenoble Alpes, CNRS, Grenoble INP, TIMC, Grenoble, France, **3** Sorbonne Université, Institute Jean Le Rond d'Alembert, UMR 7190, Paris, France, **4** Muséum national d'Histoire naturelle, UMR 7194 - Histoire naturelle de l'Homme préhistorique, Paris, France, **5** Laboratoire d'Imagerie Interventionnelle Expérimentale, CERIMED, Marseille, France, **6** Univ. Grenoble Alpes, CNRS, Grenoble INP, GIPSA-lab, Grenoble, France

* amelie.vialet@mnhn.fr

**Data Availability Statement:** All relevant data are within the manuscript.

## Abstract

As part of a long-term research project aiming at generating a biomechanical model of a fossil human tongue from a carefully designed 3D Finite Element mesh of a living human tongue, we present a computer-based method that optimally registers 3D CT images of the head and neck of the living human into similar images of another primate. We quantitatively evaluate the method on a baboon. The method generates a geometric deformation field which is used to build up a 3D Finite Element mesh of the baboon tongue. In order to assess the method's ability to generate a realistic tongue from bony structure information alone, as would be the case for fossil humans, its performance is evaluated and compared under two conditions in which different anatomical information is available: (1) combined information from soft-tissue and bony structures; (2) information from bony structures alone. An *Uncertainty Quantification* method is used to evaluate the sensitivity of the transformation to two crucial parameters, namely the resolution of the transformation grid and the weight of a smoothness constraint applied to the transformation, and to determine the best possible meshes. In both conditions the baboon tongue morphology is realistically predicted, evidencing that bony structures alone provide enough relevant information to generate soft tissue.

## Author summary

The issue of the phylogenetic emergence of speech in humans is the focus of lively and strong debates. It questions both cognitive and physical capacities of fossil hominins to articulate speech. The ultimate goal of our research project "*Origins of Speech*" is the quantitative investigation of the physical aspects of the debate. We rely for that on the design biomechanical models of fossil hominins' vocal tracts and on the assessment of their capacity to articulate distinctive sounds as is required for the emergence of spoken language. Since fossil remains do not preserve soft tissue, the technical challenge is to be able

**Funding:** This work is a production of the members of the team "Origins of speech" hosted at Sorbonne University - Institut des Sciences du Calcul et des Données. The funders had no role in study design, data collection and analysis, decision to publish, or preparation of the manuscript.

to predict them, and in particular the tongue, from bony structures alone. In this paper we present our method to reach this goal, which uses medical images of the head and neck to register a reference biomechanical tongue model of a living human into a tongue model of any other primate. We evaluate it quantitatively on the prediction of a Baboon tongue, for whom we have accurate X-Ray scans of the skull and the vocal tract, by comparing the tongue model predicted from bony structures alone with the model predicted from bony and soft tissue structures and with the tongue segmented on the baboon X-Ray data. The evaluation includes a mathematical evaluation, based on uncertainty quantification methods of the sensitivity of the predictions to the variations of crucial parameters used in the optimal geometrical registration. The results are very encouraging for future application to fossil hominins.

## Introduction

The origin of the capacity for spoken language in humans is a debated question as much in terms of the biological conditions necessary for its establishment as in terms of its emergence during human evolution. Several hypotheses have been formulated. Some of them establish a link between speech emergence, the transition from quadrupedalism to bipedalism, and the preference for manual gestures, which opened a new space of freedom but also frames for the use of the face and, then, of the mouth [1]. Alternatively some others consider this faculty as resulting from a diverted use of the masticatory apparatus, which basically were dedicated to swallowing but would also allow phonation and articulated speech by a phenomenon of exaptation [2, 3].

Concerning its emergence within human evolution, which is nearly 7 million years long and within which many genera and species have been identified, the data available are often insufficient to provide crucial insights into these debates. Fossils are by nature incomplete; the phonatory apparatus is not preserved, since it is made of cartilages and soft tissue, which do not fossilize; at the level of the oral cavity, the only remains of the tongue, which occupies most of the space, are the traces of its connection to the mandible and, rarely, the fragile hyoïd bone to which the tongue was connected.

In fossil hominins, the bones of the skull and mandible, when preserved thanks to the process of fossilization, constitute the hollow structures of the speech production apparatus. The observed anatomical arrangement on certain fossils suggests that in the first representatives of the genus *Homo* (around 2.8 Ma) a capacity for speech articulation existed, as well as the separation between the soft palate and the epiglottis (an arrangement that differs from that of great apes, where the two elements are in contact, allowing them to breathe and swallow at the same time). Moreover, at the endocranial level, although the general brain volume of such early *Homo* is small ($<600$ cm$^3$), language-specific arrangements are in place: individualized Broca's cap and a noticeable right-left asymmetry are present [4]. Does this prove though that such fossil hominins did speak like today's humans? It is not possible to answer this question, especially since it has recently been shown that some non-human primates have ranges of variation in vocal tract shapes that are similar to those observed in *Homo sapiens* [5, 6] without having developed the same type of language.

Archaeology is also called upon to prove the capacity for language in fossil hominins by considering that language mastery is expressed in material productions. Indeed, language (as practiced today by *Homo sapiens*) constitutes a means of access to abstraction, and it positions humans in their distanced relationship to the world. This is the reason why hominines

productions (*e.g.* stone tools, use of fire, habitat structuring, hunting activities) are interpreted as tangible proofs of a language capacity (even rudimentary) [7]. The symbolic manifestations, the oldest of which date back at least 500000 years and the most recent, such as the paintings on the walls of caves, are spectacular, seem to attest to a cognitive level that corresponds to that required by language [8].

Thus, both paleontology and archaeology have provided qualitative or phenomenological observations that argue for the capacity of articulated speech in fossil hominins, but neither has been able to provide measurable evidence in this direction. In this scientific context, our long-term project aims at providing a quantitative evaluation of the suitability of the biological characteristics of fossil hominins with the capacity of articulated speech. Our approach for fossil hominins in three main steps: (i) predicting from the geometry of the skull, the mandible and the vertebrae the morphology of the missing tongue, which is the organ at the core of articulated speech, and of the missing soft tissue surrounding the tongue in the oropharyngeal cavity; (ii) building a biomechanical model of the predicted fossil tongue and its surrounding structures in the oral cavity (soft palate, pharyngeal walls, hyoïd bone, lips, and mandible) including muscles that are responsible for their movements and shapings; (iii) evaluating with the biomechanical model the maximal movement magnitudes of the tongue in the anterio-posterior and the vertical dimensions, and, consequently, the range of variation of the vocal tract shapes that could be produced in these fossil hominins.

This paper constitutes a first step towards the achievement of our global project. It presents and assesses the methodology that we developed to predict tongue morphology from the available data of head and neck bony structures. This methodology extends on previous work [9] and consists in morphing a human tongue mesh, built from head and neck medical images of a reference living human, onto the anatomy of a target primate, using non-rigid registration techniques. This is done through 3D non-rigid image registration between the CT images of the reference living human subject and the CT images of the target primate. Bijar and colleagues [9] have validated a similar approach starting with the prediction of the tongue of another living human. Here, we hypothesize that such a non-rigid registration methodology is transferable into the context of tongue morphology prediction for fossil hominins, on the basis of two main observations: the variations in structural arrangement of the tongue and surrounding structures among mammals are relatively low [10], and are expected to be even lower between living and extinct humans; the non-rigid registration method adopted by Bijar and colleagues has shown fruitful results in other registration contexts with very large deformation [11, 12].

In the context of our ultimate research objective on fossil tongues, which involves the matching of head and neck morphologies that are much more different from each other than those of two living humans, the evaluation of the realism of the predicted tongue morphology is crucial. And this evaluation is a particularly complex issue, since the absence of soft tissue fossilization prevents any precise comparison between the predicted and the real tongues. To compensate for this absence, we propose in this article an evaluation of the method based on the prediction of the tongue and soft tissue of a baboon's oral cavity for which accurate 3D CT images exist for the head and neck region, making possible a quantitative comparison of prediction and data. We choose a baboon as target primate for two main reasons. First, a baboon skull shape shows strong differences with *Homo sapiens* skull shapes, but remains in the family of the skulls of the Catarrhini primates, to which hominins belong. This is a way to maximally challenge our approach. Indeed, if our methodology proves capable of capturing such strong skull shape dissimilarities, and to adequately predict the baboon's tongue and its surrounding soft tissue, we will consider that it should be able to reliably address the challenge of predicting the tongue and surrounding soft tissue of a fossil hominin, whose morphology

is less different. Second, CT images can be easily modified to simulate the absence of soft tissue, through thresholding of voxel intensity levels, which will provide a context of evaluation similar to that with fossil hominins, where only information of bony structures is available. The estimation of the baboon's tongue morphology from the corresponding skull data can then be quantitatively evaluated through comparison to the ground truth tongue shape measured from the baboon's CT exam. A specific focus concerns the uncertainty quantification of the parameters used for the non-rigid registration method. Then, the extent to which the baboon tongue shape can be reliably estimated with a registration based on bony structures alone is investigated.

## Materials and methods

### Ethics statement

CT images of the reference living human were collected in 2001 from a volunteer under informed consent. At the time, no ethical approval was required in France for this kind of data, provided that the subject was a volunteer and the physicians had agreed to carry out the acquisitions. These data have already been published in two articles in peer reviewed journals [13, 14]. CT images of the baboon were acquired on cadaver at the CERIMED (agreement No. D1305532); no ethical approval is required to use post mortem samples. This study was part of a larger program for which procedures were approved by the ethical committee on animal experimentation No. 14 (Project 68-19112012, CEEA-14 Marseille).

### Materials

Materials include, on the one hand, CT images of the head and neck of a reference living human subject and of a baboon subject, and, on the other hand, a Finite Element (FE) mesh built from structural images of the reference living human subject, called henceforth *reference human tongue mesh*.

**Structural images of human and baboon subjects.** The reference living human subject is a male adult. The set of CT images is composed of 150 axial slices 1.3 mm thick (from the middle of the forehead to the lower extremity of C3), with a resolution of 0.49 mm × 0.49 mm [13]. The baboon subject is an adult (22 years old) female Guinean baboon (*papio papio*) (frozen cadaver preserved at the Primatology Station of the CNRS, France). The cadaver was unfrozen for the CT imaging. The CT images were collected at the CERIMED (Marseille). They consist of a set of 300 axial slices 0.625 mm thick, with a resolution of 512 × 512 pixels of 1 mm × 1 mm.

**Upper surface of the baboon tongue.** The upper surface of the tongue for the baboon subject was delineated in the corresponding CT images for validation of the baboon tongue mesh prediction. A total of 608 points were located at the interface between the upper edge of tongue and the air inside the baboon oral cavity. These points were located in all sagittal slices where this tongue-air interface was clearly visible, each point being separated from its in-slice neighbors by a distance of 3 mm approximately. Fig 1 depicts the spatial distribution of these points in a sagittal slice (Fig 1(c)) and in 3D space with respect to the baboon's bony structures (Fig 1(d)).

**Paired anatomical landmarks.** A total of 21 anatomical landmarks were paired in the reference human and in the baboon images on the basis of anatomically equivalent bony structures. These landmarks were chosen for their anatomical relevance and their ease of identification in the 3D CT images. The landmarks, which correspond in their majority to the conventional landmarks defined by anatomists on dry skulls, are listed in Table 1.

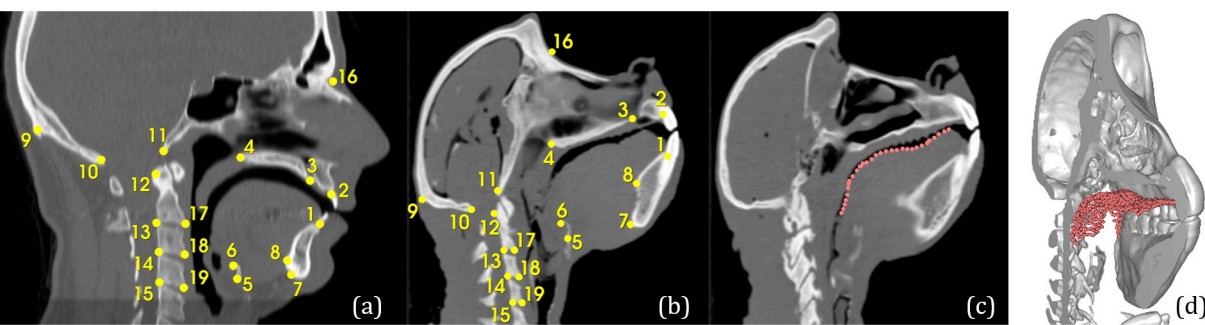

**Fig 1. 3D CT images, landmarks on bony structures and evaluation points of the upper tongue surface.** Sagittal slices for the 3D CT images of the reference human subject (a) and the baboon subject (b, c). Paired landmarks in both subjects are displayed on sagittal slices (a) and (b). The points located at the upper surface of the baboon tongue are displayed in a sagittal slice (c) and in a 3D reconstruction of the skull, the mandible and the vertebra (d).

In contrast to the points delineating the upper tongue surface described above, these anatomical landmarks were used during registration to drive the optimization problem, and not for validation. In the following, they will be denoted by $\mathcal{L}_h$ and $\mathcal{L}_b$ for the human reference and baboon subjects, respectively. Figs 1(a) and 1(b) illustrate the spatial distribution of these landmarks for the human and the baboon subjects, respectively [1].

**Table 1. Paired anatomical landmarks for the reference human and baboon subjects.**

| Landmark | Position | Conventional name |
|---|---|---|
| 1 | Cervical line of the lower incisor | *Infradental* |
| 2 | Cervical line of the upper incisor | *Prosthion* |
| 3 | Lower anterior extremity of the hard palate | Opening of the *incisive foramen* in the buccal cavity |
| 4 | Lower posterior extremity of the hard palate | *Staphylion* |
| 5 | Lower extremity of the body of the hyoïd bone | - - - |
| 6 | Upper extremity of the body of the hyoïd bone | - - - |
| 7 | Lower mental spine | - - - |
| 8 | Upper mental spine | - - - |
| 9 | External occipital prominence | *Inion* |
| 10 | Posterior border of the *foramen magnum* | *Opisthion* |
| 11 | Anterior border of the *foramen magnum* | *Basion* |
| 12 | Upper posterior extremity of the *Atlas*' Anterior Arch (C1) | - - - |
| 13 and 17 | Posterior/Anterior lower extremity of the *Atlas*' Anterior Arch (C1) | - - - |
| 14 and 18 | Posterior/Anterior upper extremity of C2's Anterior Arch | - - - |
| 15 and 19 | Posterior/Anterior lower extremity of C2's Anterior Arch | - - - |
| 16 | Between the two orbits | *Nasion* |
| 20 and 21 | Orbits: most inferior right/left points (not visible on Fig 1) | *Right* and *Left orbitale* |

All landmarks, except for landmarks 20 and 21, are placed on the mid-sagittal plane.

**Reference finite element human tongue mesh.** The reference 3D FE human tongue mesh was designed on the basis of a male subject, for whom various structural images (CT-scan, Magnetic Resonance images, X-ray scan) of the head and the oral cavity were available. The tongue mesh was generated from the tongue surface extracted from these images, where the reference subject was at rest, *i.e.* with the jaw slightly open (5 mm between the upper and lower incisor) and the tongue determining a quasi uniform cross-sectional area of the vocal tract from the glottis to the lips. We used Altair Hypermesh to generate a high quality tetrahedron mesh comprising detailed internal structures (*e.g.* 15 oriented muscle components), necessary for subsequent biomechanical simulations. The number of elements results from a compromise between the mesh complexity, the quality of its elements and the geometrical accuracy of the resulting tongue shape. As a result, the mesh contains 6039 nodes forming 29966 linear tetrahedral elements. Fig 2 depicts the FE mesh on the anatomy of the reference human subject.

## Predictions of the baboon tongue

**General principles.** Our objective in this work is to generate a FE tongue mesh of the baboon subject by morphing the FE tongue mesh of the reference human subject via non-rigid registration, while considering that only structural information from bony structures is available, as would be the case for fossil data. The accuracy of the obtained baboon tongue mesh is evaluated by comparing it with the mesh obtained from a similar registration procedure but using CT data including information on soft tissue. We consider this last mesh generation method to be a reference in terms of realism, because it takes into account structural information coming from both bony and soft tissue structures, and because previous work carried out on tongue mesh morphing between humans subjects [9] has demonstrated that under these conditions a reliable prediction of subject-specific tongue mesh can be obtained.

The mesh prediction involves a non-rigid registration problem based on two paired human and baboon CT images: one containing both soft tissue and bony structures information, and one containing information from bony structures alone, which is generated by artificially removing soft tissue information from the original CT images. For the sake of brevity and clarity, in the remaining sections we will refer to these two registration problems as *soft-tissue-based* registration and *bone-based* registration, respectively.

An overview of the proposed methodology is presented in Fig 3. It starts by solving the *soft-tissue-based* registration problem, with the aim of obtaining the best possible reference prediction of the baboon FE tongue mesh. Then, we proceed to solve the *bones-based* registration

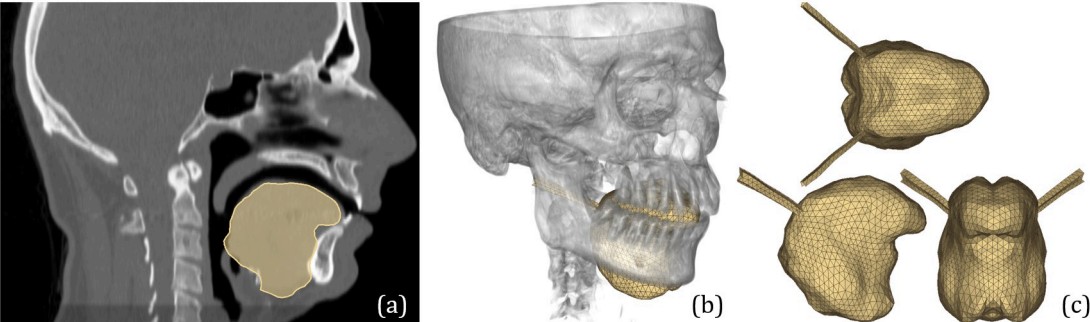

**Fig 2. FE mesh and CT image of the reference human subject.** (a) Mid-sagittal slice of the CT image with the tongue contours from the FE tongue mesh superimposed. (b) 3D reconstruction of the skull from the CT image along with FE tongue mesh inside the oral cavity. (c) Various views of the FE tongue mesh of the human reference subject.

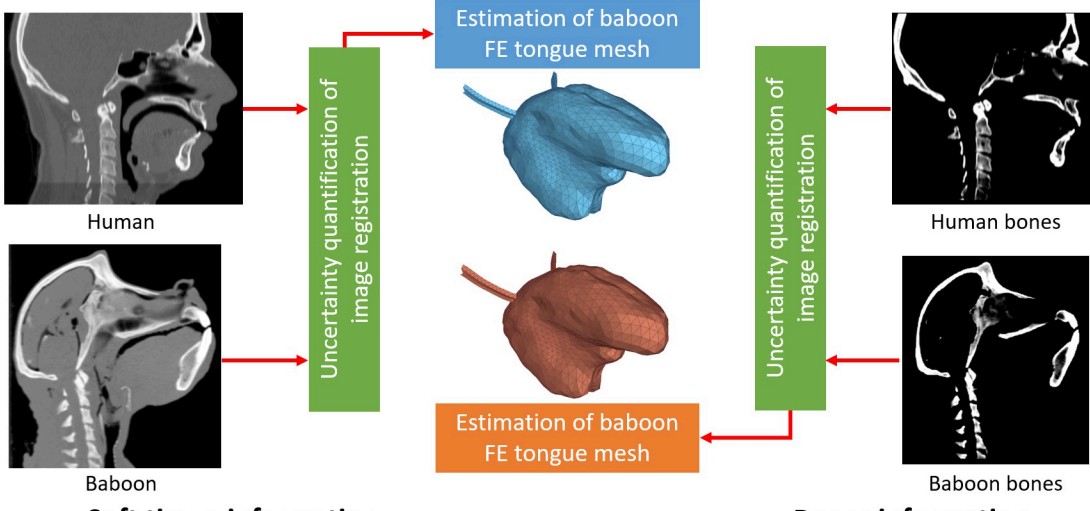

**Fig 3. Overview of proposed methodology.** Non-rigid image registration is used to estimate the tongue of the baboon subject from both the original 3D CT images and the images with bones information only. Uncertainty quantification allows the choice of registration parameters that are robust with respect to the input registration parameters.

problem. The accuracy of the FE tongue mesh predicted by the *bones-based* registration problem is evaluated by direct comparison to the prediction obtained from the *soft-tissue-based* registration problem. Finally, the uncertainty of both results to key registration parameters is quantified using a stochastic approach, as a measure of overall robustness of the image registration procedure.

**Image pre-processing.** Image pre-processing comprises a first step in which the CT images are made symmetrical with respect to the mid-sagittal plane of the head. Importantly, this first stage is not a requirement for our registration method to work efficiently. It is included in order to properly simulate the conditions in which we plan to use the method to predict human fossils' tongues. Fossil skull and mandible remains are generally incomplete, with missing pieces of bones on either side of the mid-sagittal plane. In such a case, we reconstitute a complete fossil skull-mandible set by taking into account all parts available from each side, and by symmetrizing them with respect to the mid-sagittal plane in order to reconstruct the most complete possible set. This means that we will be working *de facto* with symmetrical data sets.

In a second stage, the head and neck anatomical structures, including both soft tissue and bony structures, are segmented from the symmetrical images. These segmentations enable the estimation of image similarity during registration to be limited to regions of interest. This improves overall registration performance while avoiding unnecessary calculations. In a third and last step, only the bony structures are kept in the images in order to simulate the conditions in which the prediction of fossil hominin tongues will operate.

**Symmetrical images.** The mid-sagittal plane of the head is defined for each subject by means of three manually selected points, which describe a plane and its normal vector in 3D space with respect to which the symmetrization preserves as much as possible the global volume of the skull and the mandible. Thus, these points are not chosen for their anatomical significance but rather for their efficacy in the symmetrization process. Furthermore, this process enables to compensate for a possible tilt of the CT images with respect to the vertical plane.

Once these points are defined, the 3D CT images are translated and rotated, in such a way that (i) the thus defined mid-sagittal plane of their volume becomes aligned with the vertical plane of the cartesian working space, and (ii) the center of their volume corresponds to the origin of the working space. The thus defined mid-sagittal plane divides the cartesian working space in two parts, which are arbitrarily called *left* and *right*. The images are then symmetrized by mirroring the voxels that are located on the right with respect to the mid-sagittal plane. These symmetrical images are used as starting point for the image registration process, and will be denoted in the following by $\mathcal{I}_h$ and $\mathcal{I}_b$ for the reference human and baboon subjects, respectively.

In addition, a binary image is generated for each symmetrical image $\mathcal{I}$ by labelling each voxel as *foreground* or *background*, according to its relative position with respect to the mid-sagittal plane, *i.e.* on the right or on the left. As will be detailed in the upcoming "Image registration" section, these symmetrical binary images are used to enforce the symmetry of the estimated transformation during registration, and will be denoted by $\mathcal{S}_h$ and $\mathcal{S}_b$ for the reference human and baboon subjects, respectively.

**Head and neck region segmentation.**   Only the voxels located inside the region enclosed by the head and neck anatomical structures (and not the surrounding air) are taken into account during registration, in order to avoid any influence of voxels that do not carry any information about the morphological and anatomical differences across subjects. For this purpose, a segmentation of the head and neck region is performed using automatic thresholding, with a threshold value chosen slightly above the value corresponding to air (-1000 HU). Morphological closing and dilation operations are applied to the resulting segmented images to fill in holes inside the head and neck region, which essentially correspond to the oral and nasal cavities. As such, air-tissue interfaces are preserved inside the head and neck regions only, thus conserving information that could be valuable for the *soft-tissue-based* registration process.

**Generating images for *bone-based* registration.**   In order to simulate, in the context of the present study, the condition under which registration of fossil anatomical structures will be done (*i.e.* the *bone-based* registration problem), images of the bony structures alone are generated for each subject, by artificially removing soft-tissue information from the symmetrical images $\mathcal{I}_h$ and $\mathcal{I}_b$. This is performed automatically by replacing the intensity value of all voxels associated with soft tissue with an intensity value corresponding to air (*i.e.* -1000 HU). Since voxels associated with soft tissue have lower HU intensity values than those associated with bony structures, the latter are automatically identified by image thresholding. The hyoïd bone, which does not fossilize, is removed manually from the images. This latter processing step allows a more ascertained assessment of the registration performance to be expected when 3D CT images of fossils will be used. The resulting images of the bones are depicted on the right of Fig 3.

**Image registration.**   Let $\mathcal{I}_h : \Omega_h \mapsto \mathbb{R}, \mathcal{I}_b : \Omega_b \mapsto \mathbb{R}$ be the 3D CT symmetrical images of the reference human and baboon subjects, respectively (note that $\mathcal{I}_h$ and $\mathcal{I}_b$ refer both to the images with soft tissue information and to the images with bony structures alone, since the registration procedure is the same for both). The goal of image registration is to find a transformation $\mathcal{T} : \Omega_h \mapsto \Omega_b$, such that $\mathcal{I}_h(x)$ and the deformed $\mathcal{I}_b(\mathcal{T}(x))$ are similar under a suitable similarity criteria, for all $x \in \Omega_h$.

**Transformation model.**   For the non-rigid image registration procedure, the Free Form Deformation (FFD) transformation model with B-Spline interpolation [12] is used. With this model, the transformation of a point is given by the weighted sum of B-Spline basis functions centered at a fixed number of control points, which are uniformly distributed throughout the transformation domain. There are various motivations behind this choice. First, FFD is a

parametric transformation model, which allows a drastic reduction of the number of parameters to be determined during optimization, and thus, potentially increased robustness to local minima when compared to an approach that would determine a transformation for each individual voxel (such as diffeomorphic demons) [15]. This is specially true for the context considered in this paper, where intensity information is scarce. In addition, in comparison to other parametric models, B-Splines have a compact support, which facilitates the optimization procedure during registration as each parameter has only a local effect on the final transformation. Finally, in the medical imaging community, the FFD transformation model has been successfully used in the context of various clinical applications [11, 16], and proved to be well suited for FE tongue mesh morphing between human subjects [9]. In the following, we refer to the transformation model and the corresponding transformation parameters by $\mathcal{T}_\mu$ and $\mu$, respectively.

**Cost function.** The aim of the non-rigid image registration process is the estimation of a FE tongue mesh of the baboon subject that is suitable for numerical simulations. Towards that end, we would like the transformation $\mathcal{T}_\mu$ to respect the following criteria:

(i) Maximize the correlation of voxel intensity values between $\mathcal{I}_h(x)$ and the deformed $\mathcal{I}_b(\mathcal{T}_\mu(x))$.

(ii) Maximize the symmetry across the mid-sagittal plane of the registered images $\mathcal{I}_b(\mathcal{T}_\mu(x))$.

(iii) Minimize the distance between the registered paired anatomical landmarks.

(iv) Minimize the distortion of elements in the baboon FE tongue mesh resulting from the whole process.

We have designed a cost function that is composed of four terms. Each one of these terms has the objective of enforcing one of the four criteria listed above. This cost function, denoted $\mathcal{F}$, can be written as follows:

$$
\begin{aligned}
\mathcal{F}(\mu; \mathcal{I}_h, \mathcal{I}_b, \mathcal{S}_h, \mathcal{S}_b, \mathcal{L}_h, \mathcal{L}_b) \quad &= NC(\mu; \mathcal{I}_h, \mathcal{I}_b) + \alpha\, SD(\mu; \mathcal{S}_h, \mathcal{S}_b) \\
&+ \beta\, D(\mu; \mathcal{L}_h, \mathcal{L}_b) + \gamma\, \mathcal{P}(\mu),
\end{aligned}
\tag{1}
$$

where $NC$ corresponds to the Normalized Cross-Correlation (NCC) similarity metric, $SD$ corresponds to the Sum-of-Squared-Differences (SSD) similarity metric, $D$ is the average euclidean distance between two sets of points, $\mathcal{P}$ is a penalty term, and $\alpha$, $\beta$, $\gamma$ are the weights associated to each term.

The criteria (i), (ii) and (iii) are naturally enforced by the first three terms in $\mathcal{F}$. For the criteria (iv), we chose to penalize the bending energy:

$$
\mathcal{P}(\mu) = \frac{1}{|\Omega_h|} \sum_{x \in \Omega_h} ||H(\mathcal{T}_\mu)||_{\mathrm{F}}^2 ,
\tag{2}
$$

where $H(\mathcal{T}_\mu) \in \mathbb{R}^{3 \times 3 \times 3}$ is the Hessian matrix of $\mathcal{T}_\mu$, $|\Omega_h|$ denotes the number of elements in $\Omega_h$, and $||\cdot||_{\mathrm{F}}$ denotes the Frobenius norm. The Hessian matrix $H(\mathcal{T}_\mu)$ contains all the second partial spatial derivatives of $\mathcal{T}_\mu$. Therefore, through $\mathcal{P}$, we penalize sharp changes in local transformation curvature, enforce smoothness of $\mathcal{T}_\mu$ and minimize foldings.

In addition, to further enforce criteria (ii), the control points of the FFD $\mathcal{T}_\mu$ are distributed in alignment with the axes of $\mathcal{I}_h$ and symmetrically with respect to the mid-sagittal plane.

**Affine initialization of registration.** The affine components of the transformation $\mathcal{T}_\mu$ (*i.e.* global translation, rotation and scaling) were computed prior non-rigid registration

through an affine initialization process formulated as an optimization problem, with the average euclidean distance between a small subset of the available paired anatomical landmarks as cost function. This subset of landmarks was chosen manually to capture average size change and frontal-axis rotation of the oral cavity, and this was done using landmarks number 2, 4, 7, 20 and 21 (see "Paired anatomical landmarks"). All mentions to non-rigid registration in this paper suppose affine-initialized data.

**Assessment of registration quality.** The quality of the resulting transformation $\mathcal{T}_\mu$ is evaluated in terms of the accuracy of the predicted FE baboon tongue meshes and the quality of the elements in these meshes. Given the differences in the images considered in the process (see "Generating images for *bone-based* registration"), the results of the *soft-tissue-based* registration and the *bone-based* registration processes are expected to be different. Hence, the accuracy of their respective FE tongue mesh predictions should be computed differently. In the *soft-tissue-based* registration process, we expect the transformation $\mathcal{T}_\mu$ to map two organs that are partially visible on the 3D CT images, namely the reference human tongue and the tongue of the baboon subject. Therefore, the accuracy of the resulting transformation $\mathcal{T}_\mu$ can be computed on the basis of the proximity between the predicted baboon FE tongue mesh and the actual baboon tongue as represented in the 3D CT images. Consequently, the accuracy is calculated as the average distance between points at the actual baboon tongue surface (see Fig 1 (c) and 1(d)) and the surface of the predicted FE baboon tongue mesh. In the *bone-based* registration process, since no information is available on the actual baboon tongue, we expect the transformation $\mathcal{T}_\mu$ to produce a FE baboon tongue mesh that is as close as possible to the baboon tongue mesh predicted by the *soft-tissue-based* registration process. The accuracy is then calculated as the average node-to-node distance between the two predicted FE baboon tongue mesh surfaces: the better the prediction from the *soft-tissue-based* registration process, the better the prediction from the *bone-based* registration problem.

In addition, since the predicted FE baboon tongue meshes must be suitable for numerical simulation, the quality of their elements is also considered for the assessment of the overall prediction quality. Importantly, we have chosen to include a measure of the quality of the mesh elements as part of the optimized criterion that constrains the registration method. We could have proceeded differently, and determined first an optimal tongue transformation without constraining it by any consideration for the quality of the mesh elements before remeshing the volume of the baboon's tongue predicted by the registration procedure. We did not do so, in order to preserve the internal structure of the reference tongue mesh in which tongue muscles' anatomical implementation has been carefully determined. In addition, our approach avoid the time-consuming meshing procedure that would be a handicap in a context in which several predictions may need to be generated and assessed. More specifically, in order to preserve the high quality of the reference FE mesh of the human tongue, the predicted FE baboon tongue meshes should be devoid of inter-element penetrations and minimize distortion of the elements. The former can be achieved by ensuring that the resulting transformations $\mathcal{T}_\mu$ are diffeomorphic, *i.e.*, invertible and differentiable [17]; the latter can be indirectly enforced by controlling the amount of deformation. The quality of the predicted FE baboon tongue meshes is evaluated the same way for both registration processes. We compute the determinant of the transformation Jacobian, *i.e.* $J(x;\mu) = \det(\nabla \mathcal{T}_\mu(x))$, which should be greater than zero across the transformation domain in order to ensure diffeomorphism. The quality of the FE baboon tongue mesh predictions is assessed by computing the average of the 10% lowest $J$ values evaluated at the voxels inside the tongue in the reference image $\mathcal{I}_h$ to which the transformation applies, henceforth referred to by $J_{10\%}$.

These measures of the accuracy and quality of the predicted FE baboon tongue meshes are at the basis of the evaluation of the sensitivity of the registration processes to variations in their parameters, which leads to the selection of the best registration processes (see "Uncertainty quantification: adaptive stochastic collocation").

**Implementation and registration parameters.** The registration procedure was implemented in SimpleElastix, a python interface to the open source parametric image registration toolbox Elastix [18]. The adaptive stochastic gradient descent method is applied to minimize the cost function $\mathcal{F}$, which is estimated using 20000 points randomly sampled at each iteration from the head and neck region (see "Head and neck region segmentation"). A multi-resolution approach is adopted to cope with the large geometrical differences existing between the human reference and baboon subjects. A total of 3 spatial resolutions are used, with the resolution changing from coarse to fine by a factor 2, both for the images and the FFD grid.

The choice of registration parameters is in general not straightforward, as it greatly depends on the registration problem at hand and the expected registration accuracy. Here, we considered the alignment of the mid-sagittal planes (as assessed with the *SD* term), with respect to which the meshes have to be symmetrical, to be as important as the alignment of intensity values (*NC* term); we therefore set the value $\alpha = 1$. As for the remaining parameters, we performed a series of registrations with arbitrarily chosen values and perturbations to identify the resulting variability of qualitative registration accuracy. We obtained relatively stable registration results with the weight $\beta$ for the distance term $D$ set to 0.0, 0.05 and 0.1 for the coarsest to the finest resolution, respectively. The weight $\gamma$ of the penalty term $\mathcal{P}$ and the size of the FFD grid in the finest resolution were the two parameters resulting in the largest variation in registration accuracy. The assessment of the robustness of the image registration process to variations in these two parameters is performed through a stochastic approach entitled *adaptive stochastic collocation* described in the sequel. In the remaining of this paper, we will use the notation $\lambda = \{\lambda_1, \lambda_2\}$ with $\lambda_1$ representing the size of the FFD grid and $\lambda_2$ the weight $\gamma$ of the penalty term $\mathcal{P}$, both in the finest resolution.

## Uncertainty quantification: Adaptive stochastic collocation

The general aim of uncertainty quantification (UQ) is to quantify and analyze a model response, often called *output* or *quantity of interest* (QoI), when the model parameters are either not precisely known or are known within some probabilistic framework. The most common method to quantify uncertainties is the *Monte-Carlo* method. In this method, the main idea is to set the uncertain parameters varying within some variation intervals according to a pre-defined probability density function (pdf), for example a uniform or gaussian density distribution. Then, samples of the uncertain parameters are drawn from the uncertain parameters space and the model output is computed for each sample. Once the process has reached convergence, *i.e.* when increasing the number of samples the statistics on the model output are nearly constant, we can extract stochastic information, such as measures of the variance and mean, to assess the robustness of the model output. However, in the Monte-Carlo method reaching the convergence is very slow and it is computationally expensive, since it requires an important number of samples and computations of model outputs. For this reason, we decided to use another approach based on the *stochastic collocation method* where the samples are not fully randomly selected, but are chosen such that a stochastic error is minimized, as explained hereafter. Moreover, these samples are used next to build a *response surface* or *metamodel*, which is a continuous, interpolated surface such that one can estimate a model response as accurately as possible.

The adaptive stochastic collocation method on simplex elements proposed by Van Langenhove *et al* [19] is used here for the following reasons: (a) it is non-intrusive, meaning the image registration tools are seen as a black box and no modification of the tools is required; (b) it is an adaptive approach with a constraint on the computational cost, meaning we should get the best possible response using a computational budget; (c) the method has been built to capture singularities such as rare events or localized high sensitivity of the model response. The main idea is to estimate the stochastic error on the quantity of interest by measuring the difference between the true value of the model output and the value estimated with the metamodel using an interpolation error on the parameter space (see below for more details). This measure of the error is used to build an adapted simplex elements tessellation of the uncertain parameter space, which is essentially an adapted "grid".

**Stochastic problem formulation.**   Let us first set some mathematical notations and framework for the definition of the stochastic problem. We introduce $\Lambda \subset \mathbb{R}^n$ the space of model inputs $\boldsymbol{\lambda}$, which we refer to as the parameters of interest or uncertain parameters. Let $\mathbf{Q} : \Lambda \to D \subset \mathbb{R}^m$ denote the model response map, or quantity of interest (QoI) map, from the uncertain parameters to the space of observable model output data denoted by $D \subset \mathbb{R}^m$. We assume that $(\Lambda, \mathcal{B}_\Lambda, \mu_\Lambda)$ and $(D, \mathcal{B}_D, \mu_D)$ are two measurable spaces on some Borel $\sigma$-algebras $\mathcal{B}_\Lambda$ and $\mathcal{B}_D$ restricted to $\Lambda$ and $D$, respectively with $\mu_\Lambda$ and $\mu_D$ the associated measures on these spaces.

In this work, $\mathbf{Q}(\boldsymbol{\lambda})$ is defined as the average of node-to-node distances between two FE tongue meshes, one chosen as the best prediction for the *soft-tissue-based* registration process, and one for the prediction of *bone-based* registration process, for each uncertain parameters values $\boldsymbol{\lambda} = \{\lambda_1, \lambda_2\}$ with $\lambda_1$ representing the size of the finest FFD grid and $\lambda_2$ the weight for the penalty term. Moreover, we denote $\pi_\Lambda$ the joint probability function (pdf) associated to parameters $\boldsymbol{\lambda}$. We assume the uncertain parameters to be independent, but not necessary identically distributed.

Suppose a discretization or partition $\mathcal{H}_h$ of the parameter space $\Lambda$ into simplex elements (triangular elements here) using a finite number of samples $\boldsymbol{\lambda}_i = (\lambda_1, \lambda_2)_i, \forall i = 1, \ldots, ns$. A response surface or metamodel is built from the model response on these samples.

Then, let $\mathbf{Q}(\boldsymbol{\lambda}_i)$ be the exact stochastic response or exact quantity of interest. We aim to provide an estimate and minimize the following *stochastic interpolation error*:

$$\eta = \| \mathbf{Q}(\boldsymbol{\lambda}_i) - \Theta_h \mathbf{Q}(\boldsymbol{\lambda}_i) \|_{L^p(\Lambda)}, \tag{3}$$

where $\Theta_h$ is the (linear) interpolation operator in the parameter space such that $\Theta_h \mathbf{Q}(\boldsymbol{\lambda}_i)$ is the interpolated model response. For the purpose of this paper, we will focus on the $L^1$-norm of the interpolation error in the stochastic space. This is a purely practical choice, well adapted for approximation of potentially discontinuous solutions, but there is no restriction in using a different $p$-norm.

Since we are dealing with random quantities, we focus on minimizing the average error, which introduces the probability density function $\pi_\lambda$ into the $L^1$-error formulation:

$$\bar{\eta} = \mathbb{E}[\eta] = \int_\Lambda |\mathbf{Q}(\boldsymbol{\lambda}_i) - \Theta_h \mathbf{Q}(\boldsymbol{\lambda}_i)| \, \pi_\Lambda \, \mathrm{d}\boldsymbol{\lambda}. \tag{4}$$

We highlight here that the probability density function $\pi_\lambda$ acts as a weighting of the interpolation error. The error estimate (4) will be used as a refinement indicator to drive adaptivity over $\Lambda$, *i.e.* to decide the samples that should be drawn.

**Solving the stochastic adaptive optimization problem.**   From a mathematical point of view, we seek to solve an optimization problem where we want to find the best tessellation (*i.e.* division into triangles of the $\Lambda$ space) such that the stochastic error (4) is minimized under the

constraint of a given computational budget, defined here as the number of image registration processes we wish to use. The key idea of this method, inspired initially by physical problems, is to define and solve the optimization problem in a continuous framework where it has mathematical properties, such as convexity, that ensure the existence of a solution. We will use the continuous framework of a Riemannian metric space (RM), in which for each point (or sample) we can associate a Riemannian metric, which is a symmetrical, positive-defined matrix, that contains information regarding the direction and length of the triangular element. This framework is particularly well suited when the model response is highly sensitive to changes in the uncertain parameters considered, possibly in some privileged direction.

Using the continuous framework of a Riemannian metric space and the associated continuous error model, the stochastic error estimate (4) has the following representation in the Riemannian metric space

$$\mathbf{E}_{\boldsymbol{\lambda}}(\mathcal{M}) \;=\; \int_{\Lambda} \operatorname{trace}\!\left(\mathcal{M}^{-\frac{1}{2}}(\boldsymbol{\lambda})\,\mathbf{H}_{\mathbf{Q}}(\boldsymbol{\lambda})\,\mathcal{M}^{-\frac{1}{2}}(\boldsymbol{\lambda})\right)\mathrm{d}\boldsymbol{\lambda}, \tag{5}$$

where

$$\mathbf{H}_{\mathbf{Q}}(\boldsymbol{\lambda}) = \pi_{\Lambda} H(\mathbf{Q}(\boldsymbol{\lambda})), \tag{6}$$

with $\mathcal{M}$ a Riemannian metric, $H(\mathbf{Q}(\boldsymbol{\lambda}))$ the Hessian matrix of $\mathbf{Q}(\boldsymbol{\lambda})$ and the weight $\pi_{\Lambda}$ the probability density function defined on $\Lambda$.

The *stochastic optimization problem* is then formulated as follows:

$$\text{Find } \mathcal{M}_{\boldsymbol{\lambda}}^{opt} = \operatorname*{argmin}_{\mathcal{M}_{\boldsymbol{\lambda}}} \mathbf{E}_{\boldsymbol{\lambda}}(\mathcal{M}), \quad \text{subject to} \quad \mathcal{C}(\mathcal{M}) = \mathcal{C}_{\boldsymbol{\lambda}}, \tag{7}$$

where $\mathcal{C}_{\boldsymbol{\lambda}}$ denotes a specified complexity in the parameter space, which is equivalent to the targeted number of samples (or image registration processes) considered. The notation $\mathcal{M}_{\boldsymbol{\lambda}}^{opt}$ holds for the optimal metric that minimizes the expectation of the continuous interpolation error in the parameter space. We will use this metric to build a simplex tessellation of the parameter space, which is, roughly the "mesh" (here the "grid" nomenclature is used as a generic term for sampling and related discretization of the parameter space) associated with $\Lambda$.

The optimal stochastic metric, solution to the $n$−dimensional optimization problem (7) is

$$\mathcal{M}_{\boldsymbol{\lambda}}^{opt} = \mathcal{C}_{\boldsymbol{\lambda}}^{\frac{2}{n}}\!\left(\int_{\Lambda} \det(\pi_{\Lambda}|H(\mathbf{Q}(\boldsymbol{\lambda}))|)^{\frac{1}{2+n}}d\mu_{\boldsymbol{\lambda}}\right)^{-\frac{2}{n}} \det(\pi_{\Lambda}|H(\mathbf{Q}(\boldsymbol{\lambda}))|)^{-\frac{1}{2+n}}|\pi_{\Lambda}H(\mathbf{Q}(\boldsymbol{\lambda}))| \tag{8}$$

and the error estimate on this optimal metric is given by

$$\mathbf{E}_{\boldsymbol{\lambda}}^{opt}(\mathcal{M}_{\boldsymbol{\lambda}}^{opt}) = n\mathcal{C}_{\boldsymbol{\lambda}}^{-\frac{2}{n}}\!\left(\int_{\Lambda} \det(\pi_{\Lambda}|H(\mathbf{Q}(\boldsymbol{\lambda}))|)^{\frac{1}{2+n}}\mathrm{d}\boldsymbol{\lambda}\right)^{\frac{2+n}{n}}. \tag{9}$$

The proofs for formulations (8) and (9) are included in [19].

**Stochastic adaptive algorithm.** The previously defined optimization problem is solved iteratively and the main steps of the algorithm are outlined in Algorithm 1 described below.

**Algorithm 1** Adaptive metamodel construction and statistics computation

- Given a set of $N_{\boldsymbol{\lambda},0}$ initial samples, $\{\boldsymbol{\lambda}_i\}_0$ form the initial grid $\mathcal{H}_{\boldsymbol{\lambda},0}$ using a Delaunay triangulation.
- Run the image registration tool to compute $\mathbf{Q}(\{\boldsymbol{\lambda}_i\}_0)$.
  **For** $l$ = 1 to $n_{adap}$
  ▶ Compute optimal metric $\mathcal{M}_{\boldsymbol{\lambda},l}^{opt}$ and the stochastic error, solutions of the stochastic optimization problem (7) with given cost $\mathcal{C}_{\boldsymbol{\lambda}}$

```
              based on the QoI evaluation on the (vertices of the) previous
              grid H_{λ,l−1}.
    ▶ Use (anisotropic) information provided by the metric to generate a
      new grid H_{λ,l} containing N_{λ,l} = N_{λ,l−1} + N^{new}_{λ,l} samples.
    ▶ Use the image registration tool to compute the QoI at the N^{new}_{λ,l} new
      samples and update the metamodel Q({λ}_l) using linear interpola
      tion (stochastic collocation approach).
    ▶ If needed, compute the statistical moments of the metamodel.
  EndFor
```

We start out by (randomly) sampling the space $\Lambda$ using the probbaility density $\pi_\Lambda$ and constructing an initial grid as a Delaunay triangulation with samples $\{\lambda_i\}_0$. Here the subscript 0 holds for the initial step, *i.e.* $l = 0$, in the iterative algorithm. The image registration tool is executed for each of these samples (couple of uncertain parameters) and the QoI information $\mathbf{Q}(\lambda_i)$ is extracted. We then solve iteratively (as a fixed-point iteration) the stochastic optimisation problem (7) which has an optimal metric as a solution. The optimal metric provides information on the number of new samples $N^{new}_{\lambda,l}$ to be generated at each step $l$, as well as the position and connection of these samples with the previous triangulation. The image registration tool is evaluated on those samples to extract QoI information and the new stochastic grid is generated using an anisotropic grid generator tool FEFLO. We highlight here that the optimal metric takes into account additional information, *i.e.* the probability density function $\pi_\Lambda$. At the end of the fixed point loop, when $l = n_{adap}$, we obtain the metamodel with a computational cost controlled by the constraint.

**Stochastic parameters setup for image registration.** As stated before, we analyze the robustness of the image registration method by quantifying uncertainties on the quantity of interest $\mathbf{Q}(\lambda)$ defined as the average of node-to-node distances between two FE tongue meshes, one chosen as the best prediction of the *soft-tissue-based* registration, and one as the prediction with the *bone-based* registration, for each uncertain parameter in the $\Lambda$ space.

We study the metamodel obtained using the adaptive simplex stochastic collocation method on two uncertain parameters: (i) $\lambda_1$ defined as the grid spacing in the finest resolution; (ii) $\lambda_2$ defined as the weight $\gamma$ for the penalty term $\mathcal{P}$ in the cost function $\mathcal{F}$ defined in Eq (1). During registration, for a given $\lambda_2$ value, the $\gamma$ values used are $0.0, \frac{\lambda_2}{2}$, and $\lambda_2$ for each resolution, from coarse to fine.

The probability density functions for the uncertain parameters are:

- Gaussian probability density function $\mathcal{N}(\mu, \sigma)$ with mean (or expectation) $\mu = 20$ and standard deviation $\sigma = 4$ for the grid spacing parameter $\lambda_1$ defined on the interval [8, 32] (values expressed in millimeters).

- Uniform probability density function for the regularization weight $\lambda_2$ since we lack any *a priori* information on how this weight impacts our registration, defined on a large domain [0, 1000].

**Choice of the best predictions of the FE baboon tongue mesh.** For both registration processes, the selection of the best prediction of the FE baboon tongue mesh is based on the two measures of registration quality that are described in "Assessment of registration quality". Naturally, we seek to choose a prediction that results in high accuracy. However, accuracy alone would be an insufficient criterion as image registration is an ill-posed problem in general, and it is not likely that predictions are highly accurate but show excessive distortions. Therefore we use the Jacobian based measure $J_{10\%}$ to constrain this choice. We recall that a value of one for the determinant of the transformation Jacobian ($J(x;\mu) = 1$) implies isochoric deformation,

*i.e.* zero volume change. This could result either from the absence of deformation whatsoever, or from pure shear deformation. Considering the use of the bending energy penalty term $\mathcal{P}$ during registration, we assume that pure shear deformation remains relatively small, and thus use $J_{10\%}$ directly to control distortion.

Each prediction is assigned a score computed from the accuracy (as specified for each registration process separately "Assessment of registration quality") and Jacobian measures:

$$score = accuracy \times (J_{10\%} - 1)^2, \qquad (10)$$

and the prediction with the lowest score is selected as the best.

## Results

The uncertainty quantification procedure has enabled us to determine both for the *soft-tissue-based* and the *bone-based* registration processes the parameter values that generated the best FE baboon tongue meshes in terms of accuracy and quality, as defined in "Assessment of registration quality". The results of this procedure will be presented in details in "*Soft-tissue-based* registration" and "*Bone-based* registration". However, at first we will focus on the main goal of this work, which is to investigate whether relying on head and neck bone structures only, in the absence of any information about soft tissue, is sufficient to predict a realistic geometrical 3D representation of the tongue of a baboon subject from the mesh of the tongue of a reference human subject. To do this, we consider the best mesh obtained by the *bone-based* registration process and compare it with the best mesh obtained with the *soft-tissue-based* registration process, which we know from Bijar et al's work on humans [9] achieves a satisfactory level of realism by incorporating information on all the anatomical structures of the head and neck, whether bony or soft tissue.

### Quality of the FE baboon tongue mesh resulting from *bone-based* registration

Fig 4 presents, for a qualitative evaluation, various 2D views of the best FE baboon tongue mesh resulting from the *bone-based* registration (light blue color), superimposed to the same views of the best baboon tongue mesh resulting from the *soft-tissue-based* registration (pink color). Overall, these 2D views suggest that a strong similarity exists between the two meshes, with differences essentially located in the velopharyngeal region and in the posterior part of the tongue root. A quantitative evaluation of these differences is provided by the computation of the surface-to-surface (node-to-node) distance between the two meshes. A 3D representation of this distance is given in Fig 4(c) and 4(f), in which the colored mesh represents the prediction from the *bone-based* registration: in the largest part of the tongue mesh the distance is smaller than 0.5 mm, and its maximal value of 3 mm is reached in a relatively small part of the velopharyngeal region, on the sides of the tongue. This confirms and specifies the remarkable similarity between the two meshes.

Beyond this satisfactory result, it is interesting to note that the best accuracy for the mesh predicted by the *bone-based* registration is obtained in the front part of the tongue, *i.e.* where the tongue is the closest to the bony structures, whereas the worse accuracy is reached in the region that is located the furthest from these structures. The main explanation for this result certainly lies in the role played by the anatomical landmarks, which are all located on bony structures. Indeed, anatomical landmarks are key factors to account for large and non-homothetic global alignments between structures, which are critical in the registration of organs with large anatomical differences, as it is the case for the human and the baboon tongues. In

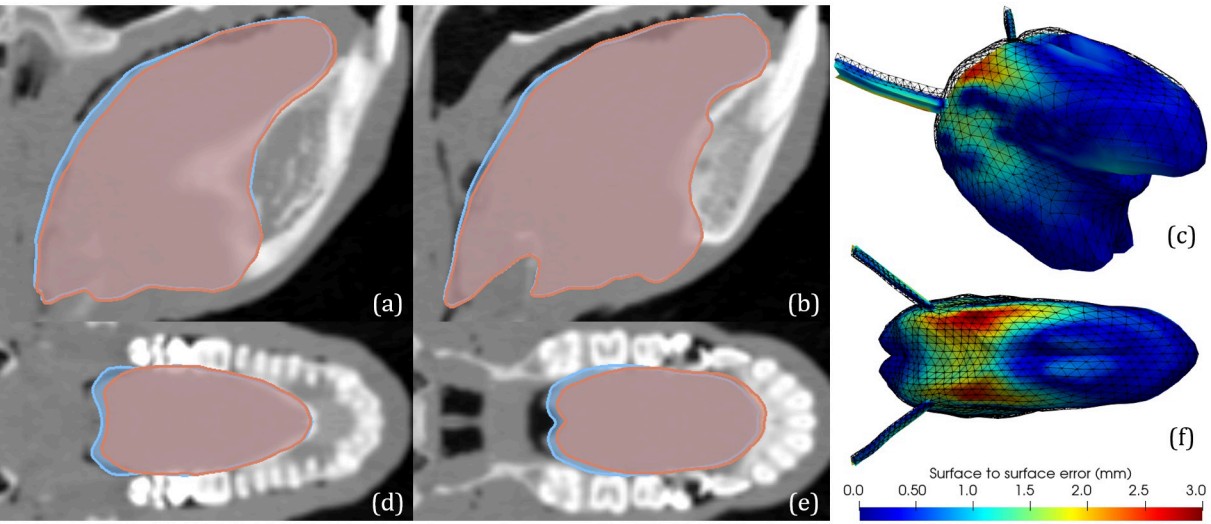

**Fig 4. Qualitative and quantitative evaluation of FE baboon tongue mesh predicted by the *bone-based* registration, considering the FE baboon tongue mesh predicted by the *soft-tissue-based* registration as the reference.** In all views, the front is on the right. Superimposed views of the meshes (pink: *soft-tissue-based* registration; blue: *bone-based* registration) on para-sagittal views at approximately 9 mm (a) and 4 mm (b) from the mid-sagittal plane, as well as on axial views at the upper mental spine (d) and at approximately 5 mm below the *infradental* (e). Perspective (c) and top (f) views of the 3D Spatial distribution of color-coded surface-to-surface distance between the two meshes (wireframe: *soft-tissue-based* registration; colored: *bone-based* registration).

the *soft-tissue-based* registration, the intensity gradient at the tongue-air interface in the oral cavity is crucial to compensate for the absence of anatomical landmarks in some regions.

The characteristics of tongue bending in the velopharyngeal region, which is at the transition between the buccal and the pharyngeal parts of the vocal tract, are largely influenced by the orientation of the head with respect to the body, which can significantly vary, independently of any intrinsic anatomical difference, with the posture of the subject in the CT-scan. This was not controlled in the collection of the baboon data, which were not specifically recorded for the purpose of the current study. In this context, in the absence of bony landmarks that inform about head orientation, a surface-to-surface distance smaller than 3 mm is a result we can be satisfied with. More generally, the statistical characteristics of the surface-to-surface distance between the two FE baboon tongue meshes (Mean value: 0.65 mm; standard deviation: 0.5 mm) describe a distance that is sufficiently small for the purposes of our work.

### *Soft-tissue-based* registration

Fig 5 depicts the results for the *soft-tissue-based* registration problem. A total of 280 uncertain parameter pairs ($\lambda_1$, $\lambda_2$) were evaluated by the uncertainty quantification method. We obtained for the predicted FE baboon tongue mesh an average accuracy ranging from 1.61 mm to 3.11 mm (Fig 5(a)), and an average $J_{10\%}$ inside the tongue volume ranging from -0.41 to 0.63 (Fig 5 (b)).

The spatial distribution of registration accuracy in the uncertainty space reveals greater uncertainty for $\lambda_1$ than for $\lambda_2$. In other words it is more likely to obtain poor registration results for a poor choice of $\lambda_1$ than $\lambda_2$, in the range of values considered. In addition, a narrow region from approximately $\lambda_1$ = 23 mm to $\lambda_1$ = 26 mm is associated with good overall accuracy, regardless of the value for $\lambda_2$. However, from the spatial distribution of $J_{10\%}$ in the uncertainty

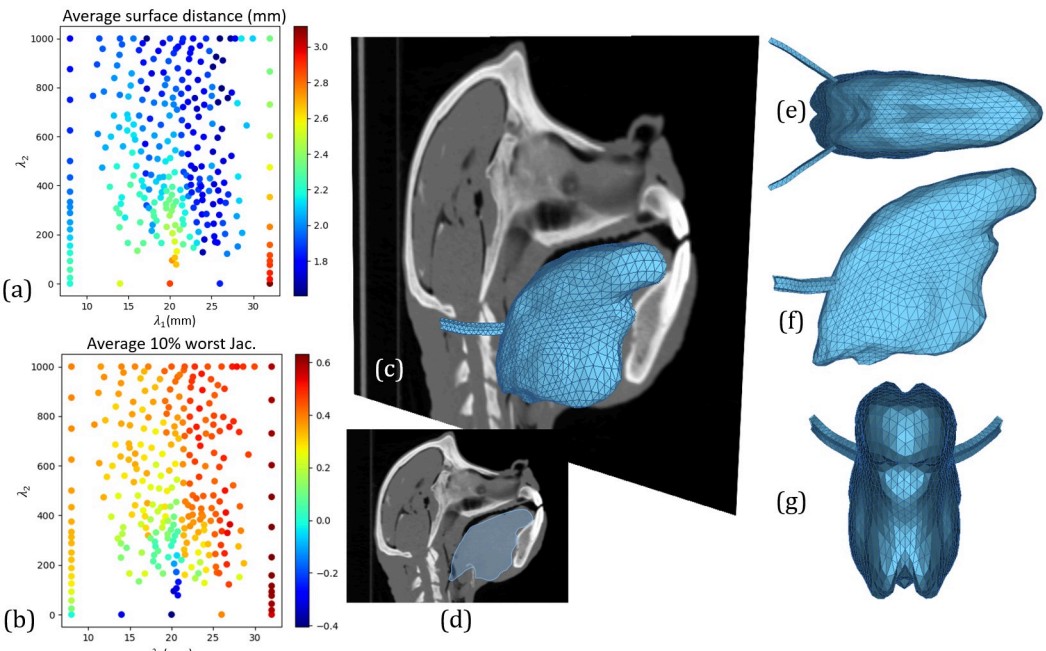

**Fig 5. Results for *soft tissue* registration problem.** Uncertain parameter space with accuracy (a) and Jacobian based measure $J_{10\%}$ (b) evaluated at 280 uncertain parameter pairs ($\lambda_1, \lambda_2$). Superposition of the baboon's mid-sagittal plane with the best predicted FE tongue mesh (c) and its mid-sagittal projection (d). Top (e), lateral (f) and frontal (g) orthogonal views of the best predicted FE tongue mesh.

space, it is clear that the higher the parameter $\lambda_2$, the less likely it is to obtain non-smooth transformations, demonstrating the usefulness of the penalty term $\mathcal{P}$.

The best result for the *soft-tissue-based* registration process is chosen as described in "Choice of the best predictions of the FE baboon tongue mesh", and it is illustrated in Fig 5 (c)–5(g). The uncertain parameters leading to this result correspond to $\lambda_1 = 26.21$ mm, $\lambda_2 = 796.74$, and correspond to an accuracy of 1.61 mm and a $J_{10\%}$ of 0.47.

### *Bone-based* registration

Fig 6 depicts the results for the *bone-based* registration process. A total of 249 uncertain parameter pairs ($\lambda_1, \lambda_2$) were evaluated by the uncertainty quantification method, resulting in an average accuracy of the FE baboon tongue mesh and a Jacobian measure $J_{10\%}$ inside the tongue volume ranging from 1.54 mm to 7.07 mm (Fig 6(a)) and -0.24 to 0.62 (Fig 6(b)), respectively.

The accuracy in the uncertainty space suggests approximately equal uncertainty for parameters $\lambda_1$ and $\lambda_2$, as can be observed in Fig 6. As opposed to the *soft-tissue-based* registration process, in the *bone-based* registration process, a poor choice of $\lambda_1$ is approximately equally likely to yield poor registration results than a poor choice of $\lambda_2$. This is due to a higher influence of the penalty term $\mathcal{P}$ in regions where soft tissue information translate in small variations of the *NC* similarity metric, thus leaving control of the transformation lonely to the remaining terms in the cost function $\mathcal{F}$.

The best registration result for the *bone-based* registration process corresponds to uncertain parameters $\lambda_1 = 25.38$ mm, $\lambda_2 = 844.35$, leading to an accuracy of 1.54 mm and a $J_{10\%}$ of 0.37.

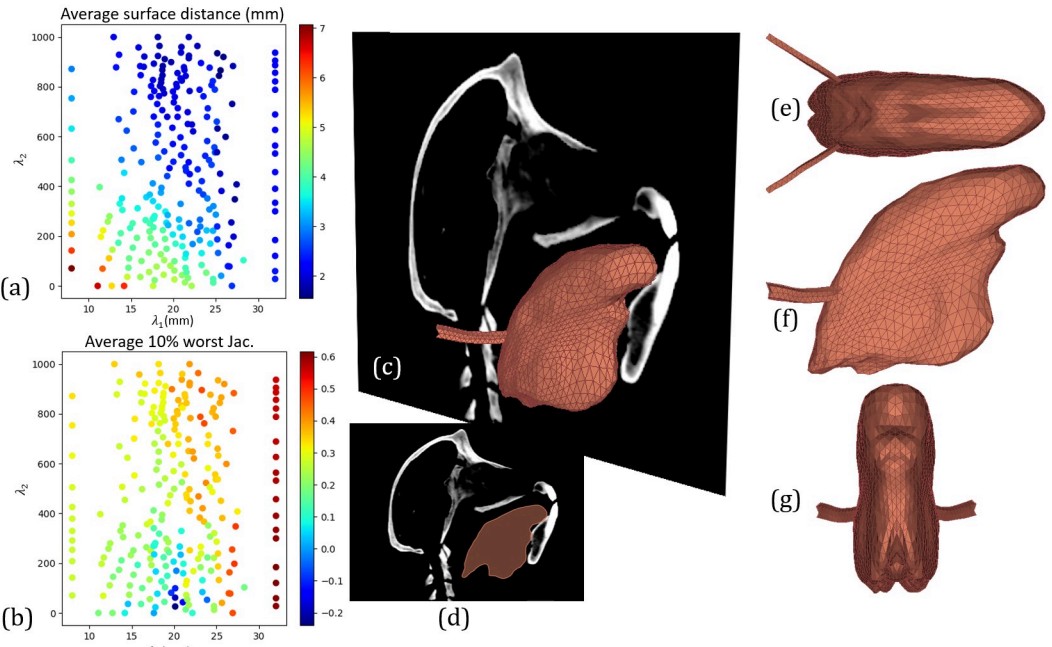

**Fig 6. *bone-based* registration process.** Uncertain parameter space with accuracy (a) and Jacobian based measure $J_{10\%}$ (b) evaluated at 280 uncertain parameter pairs $(\lambda_1, \lambda_2)$. Superposition of the baboon's mid-sagittal plane with the best predicted FE tongue mesh (c) and its mid-sagittal projection (d). Top (e), lateral (f) and frontal (g) orthogonal views of the best predicted FE tongue mesh.

It is interesting to note that these uncertain parameters are close to those resulting from the *soft-tissue-based* registration.

## Discussion

We have defined a method that registers 3D CT images of the head and neck region of a reference human subject with equivalent 3D CT images of a baboon, in order to extract a 3D geometrical deformation field that enables the anatomical structures and organs of the human subject to be projected into those of the baboon. Starting from a carefully designed biomechanical tongue model previously developed for the reference subject and its associated FE mesh [20], this 3D geometrical deformation field was applied to this reference mesh in order to estimate a FE mesh that describes the baboon tongue morphology and is suited for numerical simulations.

The morphological differences between a baboon and a living human in the head and neck region are very large, and they can only be accurately taken into account by complex, non-rigid geometrical transformations. Hence, the first aim of this work was to evaluate whether our proposed registration method was capable of estimating such a transformation by capturing the complexity of the differences in head and neck morphologies, when all anatomical structures, both soft tissue and bones, are taken into account to design the geometrical transformation field. Although there are some imperfections in the lowest posterior region of the estimated baboon tongue shape, the results obtained with the *soft-tissue-based* registration, as illustrated in Fig 5, allow us to positively answer this question. These localized imperfections are likely due to the orientation of the baboon's head, looking upwards, which generates in this region of the throat a vocal-tract bending that is clearly different from the one in our reference

subject. The proposed geometrical deformation results then from a compromise between landmark pairing and transformation regularity, but regardless, the method generates a plausible 3D mesh of a baboon tongue from the 3D mesh of a living human tongue

This work takes place in the context of our long-term project "*Origins of speech*", whose aim is to develop biomechanical models of the tongue and vocal tract of fossil hominins to quantitatively assess, via FE simulations, their ability to articulate speech. As mentioned above, only fossil data is at our disposal for the development of such biomechanical models. Our goal in the current work was therefore to first address the question: is our proposed registration method able to predict, to an anatomically plausible degree, the geometry of the tongue of a baboon subject by reducing the available information to the bony structures alone? The results obtained with the *bone-based* registration presented, in particular in Fig 4, show a remarkable similarity between the tongue meshes predicted with and without taking into account soft tissue. The bony structures of the head and neck regions seem to carry enough information about the morphological differences between living humans and baboons to reasonably well predict the tongue morphology of a baboon from the tongue morphology of a living human.

To what extent can this result be applied to the prediction of the tongue morphology of fossil hominins? Knowing that we chose to evaluate our method in the context of the transformation from a living human to a baboon, which tends to maximize the morphological differences in the head and neck region among Catarrhini primates, the Uncertainty Quantification that was carried on for both registration methods provides insights to answer this question. As explained in "Image registration", the proposed registration methods involve a *Transformation Model* that applies to a spatial FFD grid, whose spacing has to be specified, and an optimization procedure aiming at minimizing a cost function, which depends on 3 parameters $\alpha$, $\beta$, $\gamma$ (see Eq (1)). After a number of trials, we have observed that of these four factors determining the method, two have a major influence on the predictions provided by the registration: the size of the grid used in the transformation model (referred as $\lambda_1$) and the weight $\gamma$ of the penalty term $\mathcal{P}$ in Eq (1), referred as $\lambda_2$.

The results presented in Fig 6(a) show that, within the investigated ranges of the two parameters of interest $\lambda_1$ and $\lambda_2$, in the *bone-based* registration, for a given value of $\lambda_1$, the similarity between the *bone-based* and the *soft-tissue-based* predictions tends to improve when the weight of the penalty term $\lambda_2$ increases. At first glance, it is quite a surprising result, since the penalty term enforces the smoothness of the transformation, which is *a priori* not systematically compatible with a detailed approximation of the deformation. However, a more in-depth thinking suggests that, at the level of the maximal image resolution considered in this study, and for the type of information available in CT images, this is probably due to the fact that the true, ideal, geometrical transformation has intrinsic smoothness properties. Given the anatomical structures considered, and the relatively macroscopic level of description that we are interested in, this explanation sounds reasonable. This is an interesting feature of the transformation, since it makes the improvement of the accuracy compatible with the requirement to preserve the quality of the obtained meshes, in order to make possible further FE simulations. This feature certainly applies to any kind of geometrical transformation between subspecies of the Catarrhini primates, including between living and fossil hominins. In sum, our results suggest that for these geometrical transformations, in particular those associating a living human with a fossil hominin, a weight of the penalty term larger than 800 (here, and elsewhere in the paper, the values for the weights $\alpha$, $\beta$ and $\gamma$ are not normalized) ensures that the accuracy and quality of the mesh essentially depend on the size of the grid of the transformation model.

Not surprisingly for the *bone-based* registration a monotone decrease of the accuracy and of the quality is observed when the resolution of the grid of the transformation model

decreases, *i.e.*$\lambda_1$ increases (see Fig 6(a) and 6(b)). This is true for all the considered values of the weight $\lambda_2$ of the penalty term. However, since accuracy in the *bone-based* registration is evaluated in reference to the tongue mesh predicted by the *soft-tissue-based* registration, which is in turn the mesh that is evaluated with respect to the anatomical reality, it is also important to consider the impact of $\lambda_1$ on its accuracy and quality. Fig 5 shows that for this mesh, for a given value of $\lambda_2$, the influence of $\lambda_1$ on quality is not systematically monotonous, but, as could be expected, the accuracy tends to decrease when the size of the grid increases. However, it becomes monotonous when $\lambda_2$ is larger than 800 and $\lambda_1$ remains within the interval [15-30] mm. In this region of the plane $(\lambda_1, \lambda_2)$ we observed also a satisfactory level of distortion as captured by the Jacobian based measure $J_{10\%}$. Thus, all in all, in this region the results correspond to what can generally be expected for the impact of grid resolution on the quality and accuracy of the mesh prediction. Hence, it can certainly be extended to registration associating the morphology of a living human's tongue to the one of fossil hominins, which differ less from each other than living human tongues and baboon tongues.

In this transformation the mappings between the anatomical landmarks play an important role since they are major factors in the non-homothetical resizing and reorientation of the head and neck regions. Our selected landmarks have been efficient, since the mapping between the bony structures of the living human and those of the baboon is quite accurate. In the context of the prediction of fossil hominin tongues, other landmarks could be added, if necessary, to improve the accuracy of the mapping between bone structures. Importantly, in the context of the *bone-based* registration, the two landmarks located on the hyoïd bone are crucial for the prediction of the tongue morphology in the deep pharyngeal region. They indeed largely contribute to define the position of the tongue root, whose posterior part is attached to the hyoïd bone. Importantly, few hyoïd bones remain in fossil records. This absence has been at the origin of an important controversy about the height of the hyoïd bone and its influence of the phylogenetic emergence of speech [21, 22]*versus* [3, 23], which is now quite resolved [5, 6]. Today, efficient and consistent estimations of the hyoïd bone position in fossil hominins are proposed [24, 25] that enable us to estimate the position of both hyoïd bone landmarks used in our registration method from anatomical landmarks on the skull.

In this context, the proposed *bone-based* registration approach has enormous potential for the prediction of plausible biomechanical tongue models of fossil hominins from our reference human tongue model. Nonetheless, given the lack of validation data for fossil hominins, a careful evaluation of the sensitivity of the predicted tongue morphology and of its capacity to articulate speech to variations in registration inputs (*e.g.* estimated position of hyoïd bone) will be a crucial step in our approach to tackle the issue of the emergence of speech in fossil humans.

## Conclusion

This work, which quantitatively assessed the capacity of a registration method to predict a realistic morphology of a baboon tongue from the morphology of a living human's tongue, provides strong support for our ultimate objective of designing plausible biomechanical models of fossil hominin tongues embedded in their vocal tracts. Combined with an uncertainty quantification strategy similar to the one proposed in the current work, the use of these models will provide a powerful framework to assess the mobility of the fossil hominin tongues in response to muscle activations, and its sensitivity to potential inaccuracies in the estimation of crucial anatomical and mechanical characteristics, such as the position of the hyoïd bone, the stiffness of tongue tissue or the shape of the palate. We will then be able to go beyond the prediction that has been made so far only on the basis of geometrical models of the tongue [6, 21, 24, 25], which are powerful to assess how the range of variations in fossil hominin vocal tract geometry

impacts the capacity to produce distinctive sound patterns, but are limited in their capacity to reliably predict plausible tongue deformations, articulatory movement speed and the stability of potential tongue postures.

## Acknowledgments

We thank the Station de primatologie (Rousset-sur-Arc, France) for allowing us to work on a baboon specimen.

## Author Contributions

**Conceptualization:** Pablo Alvarez, Anca Belme, Yohan Payan, Pascal Perrier.

**Formal analysis:** Pablo Alvarez, Marouane El Mouss, Yohan Payan, Pascal Perrier.

**Funding acquisition:** Amélie Vialet.

**Investigation:** Pablo Alvarez, Maxime Calka, Anca Belme, Yohan Payan, Pascal Perrier.

**Methodology:** Marouane El Mouss, Anca Belme, Yohan Payan, Pascal Perrier.

**Project administration:** Amélie Vialet.

**Resources:** Gilles Berillon, Pauline Brige, Yohan Payan, Pascal Perrier, Amélie Vialet.

**Supervision:** Yohan Payan, Pascal Perrier.

**Writing – original draft:** Pablo Alvarez, Anca Belme, Yohan Payan, Pascal Perrier, Amélie Vialet.

**Writing – review & editing:** Pablo Alvarez, Yohan Payan, Pascal Perrier.

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
