## [Decision Letter · Decision Letter 0]

22 Sep 2023

Dear Dr Amélie,

Thank you very much for submitting your manuscript "Predicting primate tongue morphology based on geometrical skull matching. A first step towards an application on fossil hominins" for consideration at PLOS Computational Biology.

As with all papers reviewed by the journal, your manuscript was reviewed by members of the editorial board and by several independent reviewers. In light of the reviews (below this email), we would like to invite the resubmission of a significantly-revised version that takes into account the reviewers' comments.

Please address the skeptism from both reviewers over whether this approach would generalise to fossil hominins.

We cannot make any decision about publication until we have seen the revised manuscript and your response to the reviewers' comments. Your revised manuscript is also likely to be sent to reviewers for further evaluation.

Sincerely,

Emma Claire Robinson

Academic Editor

PLOS Computational Biology

Zhaolei Zhang

Section Editor

PLOS Computational Biology

Reviewer's Responses to Questions

**Comments to the Authors:**

Reviewer #1: The authors present a compelling case for the feasibility of predicting unknown tongue morphology by aligning 3D CT images of head/neck bone structures (serving as a substitute for fossil records) with human CT images. This research lays the groundwork for reconstructing the soft tissue of the upper vocal tract from hominin fossils, opening the door to unraveling the evolutionary biomechanics of language production. This methodology holds promise and could be of significant importance for the field of paleoanthropology. However, I have a few minor concerns regarding the approach:

1. The study employs a baboon as a surrogate for hominin variation, yet the baboon's morphology diverges considerably from that of apes, which are closer relatives to extinct hominins. For instance, the elongated muzzle of the baboon, not present in hominins, could account for much of the observed transformation. How does the performance of the method correlate with these morphological differences? One suggestion would be to assess the method's efficacy using simulated hominin-like images, generated by warping baboon or human images.

2. Could you provide more details on how the error size compares with the range of natural variability within the species? Understanding this relationship would help contextualize the study's results.

3. In studies concerning the evolution of language, the position of the hyoid bone is another critical factor to consider. How accurate is the prediction of the hyoid bone's position when using the bone-based approach? This additional metric could further validate the methodology.

Overall, the study is intriguing, but addressing these points could strengthen the implications of the work.

Reviewer #2: The study aims to predict primate tongue morphology using geometric skull matching, with a potential application to fossil hominins. Authors propose a bone-based nonlinear image registration approach since for fossil hominins, only bones of the skull and mandible are preserved. The results are satisfactory when compared with the “golden truth” of the baboon tone segmented from the CT image. The study is well structured and well written, and it’s a pleasure for me to read. Especially, the authors’ long-term goal of the project is intriguing and very interesting, aiming for a quantitative evaluation of the biological characteristics of fossil hominins with the capacity of articulated speech. However, there are some major concerns about the design of the study.

1. A major concern is the validity of the model applied to fossil hominins. While the study successfully predicts baboon tongue morphology using the chosen method, it's unclear whether this approach is valid for predicting tongues of fossil hominins. However, this assumption may oversimplify the complex relationship between skeletal and soft tissue structures, and the applicability of this model to extinct species with different craniofacial and vocal tract configurations requires further justification. Though the authors have attempted to motivate their choices (lines 110-116). I still find it difficult to be convinced. Especially authors might want to revise the following sentence: “we are confident that the proposed bone-based registration approach is powerful enough to make possible the prediction of plausible biomechanical tongue models of fossil hominins…”

2. Relating to the above is the choice of image registration method. One of the backup reasons for choosing B-spline registration is based on ref [3]. However, notice the difference of purpose of that study is different from the current one: ref[3] focuses on subject-specific FE mesh generation morphed from an atlas mesh, also between humans. Thus authors might want to revise the following sentences to acknowledge this:

This methodology is inspired from previous work [3] and consists in morphing a human tongue mesh, which was build from medical head and neck images of a reference living human, onto the anatomy of a target subject, using non rigid registration techniques.

Bijar and colleagues [3] have validated their approach starting with the prediction of the tongue of another living human.

3. Another major comment: Given that the study aims to predict morphological and geometrical aspects, the inclusion of finite element meshing may complicate the problem unnecessarily. It might be more effective to exclude the finite element meshing part to focus solely on geometry/morphology. Especially, since meshing a tongue to FE mesh would be easy using e.g. Hypermesh as authors are familiar with especially if authors choose to mesh with tetrahedral elements.

4. The choice of image registration approach is essential to this task. Authors should consider more registration methods besides the current B-Spline, e.g. Diffeomorphic demons, or other similar. especially one reason for this choice is based on ref3, while mentioned above, that study had a different focus.

5. Related to the above, it would be valuable to explore how different registration methods contribute to uncertainties, supplementing the current uncertainty quantification of parameters.

6. Consider adding a fossil hominin skull bone to the study to complete the story and demonstrate the result of a predicted tongue for a fossil hominin.

7. If the authors chose to focus on the geometry aspect, and due to the motivation in lines 110-115, would it benefit to morph the baboon skull/tongue directly to a fossil hominin instead of morphing from a human?

8. Ethical approval statement is lacking, especially since this study involves CT scans of a human and baboon.

Specific comments

1. Figure 2 seems to show the CT image of the human misses the upper part. Please clarify if this is just an illustration problem or indeed so.

2. Figure 5(d) seems the predicted tongue covers the void (throat region?) please clarify.

3. Could authors consider showing an image showing the segmented tongue of the human subject and the baboon? Seems rather difficult to see the geometry of the tongue of the baboon Figure 1(c), and is only the upper surface identifiable? Also, a related question, seems the tongue would deform easily, how would this compound the problem? E.g. necessary to have the human subj tongue at a similar position as a baboon (one would imagine it difficult to instruct baboon to do – forgive if this is too naïve question)

**Have the authors made all data and (if applicable) computational code underlying the findings in their manuscript fully available?**

Reviewer #1: **No: **The link to the data file is broken.

Reviewer #2: Yes

PLOS authors have the option to publish the peer review history of their article (what does this mean?). If published, this will include your full peer review and any attached files.

Reviewer #1: No

Reviewer #2: No
---

## [Decision Letter · Decision Letter 1]

8 Jan 2024

Dear Dr Amélie,

We are pleased to inform you that your manuscript 'Predicting primate tongue morphology based on geometrical skull matching. A first step towards an application on fossil hominins' has been provisionally accepted for publication in PLOS Computational Biology.

Best regards,

Zhaolei Zhang

Section Editor

PLOS Computational Biology

Zhaolei Zhang

Section Editor

PLOS Computational Biology

Reviewer's Responses to Questions

**Comments to the Authors:**

Reviewer #1: The authors have satisfactorily addressed my concerns.

Reviewer #2: The authors have addressed well the comments from my side and I have no further to add. Congrats on a nice study!

**Have the authors made all data and (if applicable) computational code underlying the findings in their manuscript fully available?**

Reviewer #1: Yes

Reviewer #2: Yes

PLOS authors have the option to publish the peer review history of their article (what does this mean?). If published, this will include your full peer review and any attached files.

Reviewer #1: No

Reviewer #2: **Yes: **Xiaogai Li

---

## [Editor Report · Acceptance letter]

17 Jan 2024

PCOMPBIOL-D-23-01170R1 

Predicting primate tongue morphology based on geometrical skull matching. A first step towards an application on fossil hominins

Dear Dr Vialet,

I am pleased to inform you that your manuscript has been formally accepted for publication in PLOS Computational Biology. Your manuscript is now with our production department and you will be notified of the publication date in due course.

With kind regards,

Anita Estes
